# Flogomicina: A Natural Antioxidant Mixture as an Alternative Strategy to Reduce Biofilm Formation

**DOI:** 10.3390/life13041005

**Published:** 2023-04-13

**Authors:** Chiara Amante, Chiara De Soricellis, Gianni Luccheo, Luigi Luccheo, Paola Russo, Rita Patrizia Aquino, Pasquale Del Gaudio

**Affiliations:** 1Department of Pharmacy, University of Salerno, Via Giovanni Paolo II, 132, 84084 Fisciano, SA, Italy; camante@unisa.it (C.A.);; 2Anvest Health s.r.l., Via Rosario Livatino, 84083 Castel San Giorgio, SA, Italy

**Keywords:** N-acetylcysteine, bromelain, ascorbic acid, Ribes nigrum, resveratrol, pelargonium, antimicrobial activity, anti-biofilm activity, antiadhesion property

## Abstract

The National Institute of Health has reported that approximately 80% of chronic infections are associated with biofilms, which are indicated as one of the main reasons for bacteria’s resistance to antimicrobial agents. Several studies have revealed the role of N-acetylcysteine (NAC), in reducing biofilm formation induced by different microorganisms. A novel mixture made up of NAC and different natural ingredients (bromelain, ascorbic acid, Ribes nigrum, resveratrol, and pelargonium) has been developed in order to obtain a pool of antioxidants as an alternative strategy for biofilm reduction. The study has demonstrated that the mixture is able to significantly enhance NAC activity against different Gram-positive and Gram-negative bacteria. It has shown an increase in NAC permeation in vitro through an artificial fluid, moving from 2.5 to 8 μg/cm^2^ after 30 min and from 4.4 to 21.6 μg/cm^2^ after 180 min, and exhibiting a strongly fibrinolytic activity compared to the single components of the mixture. Moreover, this novel mixture has exhibited an antibiofilm activity against *S aureus* and the ability to reduce *S. aureus* growth by more than 20% in a time-killing assay, while on *E. coli*, and *P. mirabilis*, the growth was reduced by more than 80% compared to NAC. The flogomicina mixture has also been proven capable of reducing bacterial adhesion to abiotic surfaces of *E.coli*, by more than 11% concerning only the NAC. In combination with amoxicillin, it has been shown to significantly increase the drug’s effectiveness after 14 days, offering a safe and natural way to reduce the daily dosage of antibiotics in prolonged therapies and consequently, reduce antibiotic resistance.

## 1. Introduction

The National Institute of Health has reported that approximately 80% of chronic infections are associated with biofilms, and the Center for Disease Control and Prevention (CDC) has reported that biofilm formation is present in about 65% of bacterial infections [1]. Biofilm-related illnesses, such as respiratory infections or urinary tract infections caused by the prolonged use of catheters and urethral stents/sphincters [2,3], are typically permanent and difficult to eradicate, especially in the elderly and patients who are immunocompromised or live with co-morbidities [4]. Several bacteria such as *Pseudomonas aeruginosa*, *Staphylococcus epidermidis*, *Escherichia coli*, *Proteus mirabilis*, and *Staphylococcus aureus* can form a biofilm, thereby increasing the risk of chronic infections including prostatitis and dental caries, diabetic feet and cystic fibrosis infections; they also compromise medical devices and catheters [5]. Biofilms protect the microorganism not only from harsh conditions such as pH or osmolarity, but also from antibiotics and immune cells [6]. Moreover, during the formation of the biofilm, microorganisms can produce ROS, important intra, and extracellular signaling molecules to maintain a redox cycle essential to promote microbial attachment. When microorganisms are exposed to antimicrobial agents or environmental stresses, the accumulation of ROS can result in a phenomenon recognized as oxidative stress, to which microorganisms respond by activating their redox defense mechanisms [7]. Therefore, an interesting strategy to inhibit biofilm formation and reduce antibiotic resistance could be the use of non-antibiotic therapies such as antioxidants. These compounds donate electrons or transfer hydrogen atoms to stabilize or/and terminate the ROS chain reaction, and could play an important role in the regulation of biofilm formation in both Gram-positive and Gram-negative bacterial cells [7].

Several studies have revealed the potential role of N-acetylcysteine (NAC), a thiol-containing antioxidant used as a precursor of glutathione synthesis [8], in reducing biofilm formation induced by different microorganisms [9,10,11]. It is able to reduce the production of the extracellular polysaccharide matrix and of mucus viscoelasticity thanks to the thiol group that disrupts the disulfide bonds in mucus [12], thereby increasing the permeability of antibiotics and encouraging the eradication of mature biofilms [13].

Bromelain, a crude extract of the pineapple (*Ananas comosus*), is a mixture of endopeptidases, carbohydrates, and glycoproteins [14]. Although bromelain has known anti-inflammatory and necrotic tissue debridement properties, due to its capability to hydrolyze some peptide bonds in the bacterial cell wall, it could be used against opportunistic pathogens [15].

Ascorbic acid, an antioxidant agent involved in human cell-mediated immunity [16], has been reported to be a promising antibacterial agent both alone and in combination with other antibacterial substances against different pathogens such as *S. aureus* and *P. aeruginosa* [17,18].

Blackcurrants (*Ribes nigrum* L.), compared with other berries such as strawberries and raspberries, are considered a rich source of polyphenol compounds such as anthocyanins, phenolic acid derivatives, and flavonols [19]. Various authors have recently shown the antibacterial action of blackcurrant extracts against Gram-negative and Gram-positive bacteria [20].

Resveratrol (3,5,4′-trihydroxystilbene), a natural polyphenolic antioxidant product belonging to the stilbene family, is a major component of red wine. It is known for its wide range of biological effects including antioxidant, anti-microbial, anti-inflammatory, anti-aging, anti-carcinogenic, cardio-protective, and neuroprotective properties [21,22]. Several studies have described the anti-microbial properties of resveratrol in reducing the virulence of some bacteria such as uropathogenic *E. coli* [23], and *S. aureus* [24].

Pelargonium, belonging to the Geraniaceae family, is widely used for intestinal problems, wounds, and respiratory ailments. Moreover, in the form of essential oil and in combination with NAC, it has shown antibacterial and antifungal activity [25,26].

Therefore, this work aimed to combine a pool of natural antioxidants in a novel mixture named flogomicina as an alternative strategy to reduce antibiotic resistance. In detail, flogomicina is composed of NAC, bromelain, ascorbic acid, Ribes nigrum, resveratrol, and pelargonium was tested to evaluate its antimicrobial activity, favoring the breakdown of the bacterial biofilm and the reduction of bacterial adhesion on abiotic surfaces.

## 2. Materials and Methods

N-acetylcysteine (NAC), bromelain, ascorbic acid, Ribes nigrum, resveratrol, and pelargonium were kindly donated by Anvest Health s.r.l. (Castel San Giorgio, Salerno, Italy). Deoxyribonucleic acid (DNA), mucin from the porcine stomach, diethylenetriaminepentaacetic acid (DTPA), RPMI 1640 amino acids solution, egg yolk emulsion, sodium chloride, and potassium chloride, used for artificial mucus preparation, and barbital buffer were purchased from Sigma-Aldrich (Milan, Italy), while the hydroxyethylcellulose was obtained from A.C.E.F. SpA (Piacenza, Italy). Mueller–Hinton broth (MHB) and Mueller–Hinton agar (MHA) were acquired from Biolife (Milan, Italy); trypticase soy broth (TSB) was obtained from Difco Laboratories, and phosphate-buffered saline (PBS) from GIBCO, Thermo Fisher Scientific. All other solvents and chemicals were of analytical grade.

### 2.1. Preparation of Flogomicina Mixture

After a series of pilot experiments, flogomicina (302021000020324), belonging to the category of food supplements, was prepared by combining different natural components (as reported in Table 1) in specific amounts, using a geometric dilution technique to obtain a homogeneous physical mixture

### 2.2. Permeation Experiments in Viscous Artificial Fluid

#### 2.2.1. Preparation of Artificial Fluid

Artificial fluid (AF) was prepared based on the preparation of artificial mucus reported in previous work [27] by dissolving hydroxyethylcellulose as an inert thickening agent to obtain a specific viscosity. Briefly, 15 g of hydroxyethylcellulose was dissolved in distilled water and heated under magnetic stirring to obtain a homogeneous solution. After cooling to room temperature, 4 g of DNA, 5 g of mucin, 5.9 mg of DTPA, 20.0 mL of RPMI 1640 amino acid solution, 5 mL of egg yolk emulsion, 5 g of sodium chloride, and 2.2 g of potassium chloride were added and mixed in a final volume of 1 L. After adjusting the pH value to 7.0 with NaOH, the solution was left to equilibrate at 25 °C for 12 h before any analysis.

#### 2.2.2. Permeation Assay

The penetration ability of formulation through AF was investigated by an in vitro permeation experiment using Franz-type vertical diffusion cells (Hanson Research Corporation, Chatsworth, CA, USA) with an exposed surface area of 0.6 cm^2^ [28]. The donor compartment consisted of a thin layer (3 mm) of the AF, and the acceptor compartment was filled with 5 mL of distilled water. The solution was thermostated at 37 °C by recirculating water from a thermostatically controlled bath, and magnetically stirred at 150 rpm by Teflon-coated stirring bars placed in the receptor compartment to prevent boundary layer effects. A nitrocellulose membrane (25 mm diameter and 0.45 µm pore size, Merk KGaA, Darmstadt, Germany) was applied between donor and receptor compartments. About 25 mg of the selected powder, precisely weighed, was uniformly spread on the fluid layer, and at defined time intervals between 5 and 180 min, 200 µL of samples were collected from the acceptor compartment, and the aliquots withdrawn were replaced by the same volume of water. The amount of NAC permeated was analyzed by HPLC analysis. Briefly, NAC raw material was solubilized in a CH_3_CN/H_2_O solution (1:1 *v*/*v*); then, the sample dilutions were analyzed using the same HPLC-DAD method (Agilent 1100 series system equipped with a Model G-1312 pump, a Rheodyne Model G-1322A loop (50 μL), a DAD G-1315 detector, Agilent Technologies, Santa Clara, CA, US) in order to obtain a calibration curve. The analytical mobile phase consisted of CH_3_CN and phosphate buffer 50 mM (37/63 *v*/*v*), adjusted to pH 4.50 (±0.05) with a few drops of phosphoric acid 1% (*v*/*v*). A linear response over the concentration range of 5–50 μg/mL was obtained. The UV detection was performed at 205 nm. All results were expressed as the quantity of NAC permeated per permeation area (mg/cm^2^) related to time. A peak associated with the NAC in the flogomicina mixture was identified by its retention time and confirmed by co-injection with a standard, as well as by a UV spectrum compared with the standard. Each analysis was performed in triplicate, and results were expressed as mean ± standard deviation.

### 2.3. Bacteria

The antibacterial efficacy of the formulation was tested on Gram-positive *Staphylococcus aureus* and Gram-negative *Pseudomonas aeruginosa*, *Escherichia coli*, and *Proteus mirabilis*. *Staphylococcus aureus* (ATCC 6538 and A170) and *Pseudomonas aeruginosa* (ATCC 27853) were obtained from LGC Standards S.r.L. (Milan, Italy), while *Escherichia coli* (ATCC DH5a) and *Proteus mirabilis* (ATCC 51286), a strain isolated from patients affected by a urinary infection, were purchased from Chromachemie (Basel, Switzerland).

### 2.4. Disc Diffusion Assay

A disc diffusion assay was carried out on MHA according to the Clinical and Laboratory Standard Institute (CLSI) guidelines, with suitable modifications [29]. Bacteria colonies freshly grown in MHA were inoculated into MHB and incubated overnight at 37 °C, under aerobic conditions. A sterile cotton swab was dipped in the bacterial suspension of *S. aureus* or *P. aeruginosa* normalized to 0.8 × 10^8^ cells/mL and then spread on an MHA agar plate. The plate was spotted with 15 mg of flogomicina corresponding to about 5 mg of NAC, using NAC diluted with lactose and pure NAC as positive controls. After incubation at 35 °C overnight, the diameters of the clearing zone, representing the inhibition zone around the powders, were compared using Image J software. Experiments were conducted in triplicate.

### 2.5. Biofilm Formation Assay

To assess the ability of NAC and formulation to prevent bacterial biofilm formation [30], *S. aureus* 6538 was grown overnight in an MHB medium. Overnight cultures were diluted to approximately 2 × 10^6^ cells/mL in MHB and transferred to a 96-well tissue-culture-coated polystyrene flat-bottomed plate (200 µL/well). Formulations were properly dissolved in DMSO to test increasing amounts of NAC in bacterial culture: 0.9, 4.5, 180, 360, and 720 μg/mL. Untreated cells were used as a control to confirm the successful formation of biofilm, while pure and lactose-diluted NAC powders were used as positive controls. After 48 h of incubation at 37 °C without shaking, the culture medium was removed from each well by pipetting and the plate wells were washed twice with PBS to remove unattached cells. Afterward, biofilms were dried at 37 °C for 1 h and stained with 3% (*w*/*v*) of crystal violet (CV) solution. After rinsing with water, to quantify the biofilms mass, the CV was solubilized with 200 μL of 33% acetic acid and absorbance was measured at 600 nm using a microplate spectrophotometer (Thermo Scientific Multiskan Spectrum Model #1500, Thermo Scientific, Milan, Italy).

Each assay was performed in triplicate on separate days.

### 2.6. Fibrinolytic Effect

Fibrinolytic activity was determined by a fibrin plate assay as follows. The fibrinogen solution (4 mL of 5 mg/mL fibrinogen in 0.1 mol/L barbital buffer) at pH 7.75 was mixed with agarose solution (4 mL of 20 mg/mL agarose in barbital buffer) and 0.2 mL of 20 NIH (the standard unit for thrombin) U/mL thrombin. The mixture was poured into a Petri dish and left for 1 h at room temperature. Enzymatic preparation precise aliquots were placed on the fibrin clot and incubated for 12 h at 37 °C in a moist incubator. Different concentrations of each enzymatic preparation were assayed in three independent experiments, and data were plotted as a double log. The results were compared with plasmin as the positive control.

### 2.7. Time-Killing Assay

Time-killing studies were performed on *S. aureus* strain A170. Briefly, a culture was incubated at 35 °C for 24 h. Then, bacterial suspension (0.5 McFarland scale) in MHB, corresponding to 1.5 × 10^6^ CFU/mL (colony forming unit per mL) was prepared and incubated at 35 °C under constant shaking. 200 µL of bacteria suspension was used to inoculate sterile 96-well plates. After 24 h, bacteria cells were treated with 0.01 mg/mL of amoxicillin, 0.3 mg/mL of flogomicina, and different components of flogomicina dosed in the same amounts contained in the mixture. A saline solution was used as a positive control. At defined time points (1, 3, 5, 7, 14, and 21 days) viable counts were determined to count CFU/mL by plating serial dilutions on MHA incubated at 35 °C. Kill curves were plotted with time against the number of CFU recovered.

Time-killing studies were also performed on *E. coli* (ATCC DH5a) and *P. mirabilis* (ATCC 51286). Briefly, the bacterial culture was incubated at 37 °C for 24 h. Then, bacterial suspension (0.5 McFarland scale) in MHB, corresponding to 1.0 × 10^6^ CFU/mL (colony forming unit per mL) was prepared and incubated at 37 °C under constant shaking. A total of 200 µL of the bacteria suspension was used to inoculate sterile 96-well plates; these were then treated with either 0.3 mg/mL of flogomicina mixture, NAC dosed in the same amounts contained in the mixture, or saline solution used as a positive control. At defined time points (1, 2 and 3 days) viable counts were determined to count CFU/mL by plating serial dilutions on MHA incubated at 37 °C. Kill curves were plotted with time against the number of CFU recovered. Each antimicrobial assay was performed in triplicate on separate days.

### 2.8. Anti-Adherence Effect on Abiotic Surfaces

*S. aureus* (ATCC 6538), *E. coli* (ATCC DH5a), or *P. mirabilis* (ATCC 51286) were streaked on MHA and then incubated at 25 °C for 48 h. Vital cells (for each strain) were transferred to TSB containing 0.9% D-glucose and kept at 25 °C for 24 h. Cells were centrifuged and washed twice with 0.5 mL of PBS and suspended in 1 mL of TSB and adjusted to 1.0 as a final OD600 (Optical density) nm value. Some 100 μL of this suspension was inoculated into individual wells of polystyrene 96-well plates, using TSB as a negative control. After that, the plates were incubated at 25 °C for 90 min (adhesion period). PBS was used to gently wash and remove any non-adherent cells. Then, 100 μL of fresh TSB containing the flogomicina mixture and NAC dosed in the same amounts contained in the mixture was added to the wells. After 24 h, liquid media containing the non-adherent bacteria were discarded through two rounds of washing with 200 μL of PBS. Adherent cells to the plastic surfaces were quantified using a crystal violet assay. The experiment was performed in triplicate.

### 2.9. SEM Analyses of Biofilm

Scanning electron microscope (SEM) analyses were conducted to visualize the structure of biofilms on the aluminum substrate either untreated or treated with flogomicina. The protocol for microbial biofilm sample preparation for SEM analysis consisted of gently washing with PBS the samples and a dehydration/fixing procedure of the microorganisms with increasing ethanol series concentrations (from 10 to 95%, step 10%) for 15 min for each step [31]. Afterward, samples were coated with a gold layer (100–200 Å thickness) and analyzed using a Carl Zeiss EVO MA 10 microscope with a secondary electron detector (Carl Zeiss SMT Ltd., Cambridge, UK). Analyses were conducted at 10 keV.

### 2.10. Statistical Analyses

The measurements of all analyses were carried out at least in triplicate for each sample, and the results were reported as the mean of the individual values. Statistical analysis was performed using GraphPad Prism 9.04 (Boston, MA CA, USA). The statistical test used was a two-way ANOVA followed by Tukey’s post-test or a one-way ANOVA followed by Bonferroni’s post-test, or the Mann–Whitney test where appropriate. The *p* values are indicated in the individual graphs. The results are expressed as means ± SD. Data were statistically analyzed using an unpaired Student’s *t*-test. A *p*-value of <0.05 was considered statistically significant.

## 3. Results

### 3.1. Permeation Study

With the aim of investigating the ability of pure NAC, in comparison with NAC released from flogomicina, to permeate through an artificial fluid (AF), in vitro permeation studies were conducted using Franz-type vertical diffusion cells. Comparing the permeation profile of the formulation with only NAC (Figure 1), it is possible to notice that the permeation of NAC released from flogomicina moved from 2.5 to 8 µg/cm^2^ after 30 min, and from 4.4 to 21.6 µg/cm^2^ after 180 min, suggesting an increased ability to pass through the viscous fluid related to the presence of other components in the mixture.

### 3.2. Antimicrobial Activity

A preliminary antimicrobial test against Gram-positive (*S. aureus*) and Gram-negative (*P. aeruginosa*), using a modified disc diffusion assay (Figure 2), revealed the antibacterial activity of the flogomicina containing 5 mg of NAC. For both *S. aureus* and *P. aeruginosa,* an inhibition zone of 35% and 15% was caused by flogomicina, when compared to pure NAC, respectively. The test also showed a slight increase in the inhibition halo for the NAC diluted with lactose, probably due to the better dissolution and spreading of the NAC powder on the plate.

Based on these results, the antibiofilm activity of the flogomicina mixture was carried out on *S. aureus*. Therefore, mature biofilms grown for 48 h were exposed to flogomicina containing increasing concentrations of NAC (from 0.9 to 720 µg/mL) in comparison with NAC raw material and NAC mixed with lactose, to measure their antibiofilm activity. Cell counts after treatment were determined using OD at 600 nm. NAC treatment did not result in consistently lower cell counts; on the contrary, exposure to flogomicina showed (with the only exception being for the minimum tested concentration (0.9 µg/mL)) a significant decrease in biofilm formation for all tested concentrations, as shown in Figure 3.

### 3.3. Fibrinolytic Activity

Since fibrin plays an important role in biofilm formation by different bacteria, as in the case of *S. aureus* [32], the fibrinolytic effect of both the bromelain and flogomicina mixtures was evaluated by performing a fibrin plate assay. The results of the fibrinolytic activity have been reported versus caseinolytic activity in a double-log scale showing a linear correlation for bromelain as well as for flogomicina (Figure 4). Interestingly, flogomicina exhibited a stronger fibrinolytic activity compared to plasmin (an endogenous serine protease that degrades fibrin blood clots) and bromelain alone, especially when lower concentrations were tested.

### 3.4. Time-Killing Assay against S. aureus, E. coli, and P. mirabilis

The ability of the formulation to prevent microbial growth was tested by performing a time-killing assay against *S. aureus*, *E. coli*, and *P. mirabilis*. In Figure 5, the activity against *S. aureus* was reported, comparing flogomicina with only NAC, and a mixture composed of pelargonium, ascorbic acid, and bromelain were all administered three times a day. Treatment with flogomicina showed a statistically significant increase in the power to inhibit bacterial growth time-dependently, compared to a single component tested in the same quantity present in flogomicina. This trend is strongly evident after the fifth day of monitoring, when the treatment with the multi-component presented a noticeable action about 20% greater than the treatment of NAC alone, and 55% greater than the mixture containing pelargonium, ascorbic acid, and resveratrol.

Based on these results, in order to verify the ability of flogomicina to enhance the action of amoxicillin working as an adjuvant over antibacterial treatments, *S. aureus* was treated with amoxicillin (0.01 mg/mL), flogomicina (0.3 mg/mL) and a blend of the two. Figure 6 showed a relevant time-dependent potentiation of amoxicillin activity after 14 and 21 days, thanks to the co-administration three times a day of flogomicina, supporting the notion of synergy between amoxicillin and NAC.

As shown in Figure 7, 0.3 mg/mL of flogomicina reduced around 80% of *E. coli* growth and *P. mirabilis* after just 1 day, showing more efficiency than only NAC, which reduced the bacterial growth by 60% at the same time.

### 3.5. Anti-Bacterial Adhesion

Figure 8 shows that both N-acetyl cysteine and flogomicina had a higher inhibitory effect on the adherence of *S. aureus* and *P. mirabilis* in comparison to *E. coli*. Specifically, flogomicina showed a higher effect (*p* < 0.05) than NAC on all the tested bacteria adherence, with a higher difference in the case of *E. coli*, where flogomicina was 11% more active than NAC.

## 4. Discussion

The biofilm is a structured multicellular microbial community of microorganisms, enclosed in a self-produced polymeric matrix and capable of adhering to the biological surface or abiotic material, becoming responsible for infections in different organs and devices [33]. Biofilm-based infections are generally treated with antibiotics, which represent the traditional pharmacological approach. However, in some cases, they are unsuccessful due to their slow or partial penetration into the deepest layers of biofilm [13]. In this work, flogomicina, a marketed natural mixture of antioxidants including NAC, was prepared and tested in order to determine whether the antimicrobial effect of the mixture was greater than the solo NAC widely used as an alternative pharmacological approach to control bacterial biofilm growth in human diseases [34,35,36].

NAC’s activity has been principally attributed to its disaggregating properties due to the presence of the thiol group that promotes cleavage of the disulfide bonds that are prerequisites for the stability of the extracellular substances in microbial biofilms [35]. Figure 1 reported the permeation profile of pure NAC in parallel with NAC present in flogomicina. The results show that the combination of NAC in a complex mixture allowed a more efficient permeation through the artificial fluid. This effect could be related to the presence of bromelain in the flogomicina mixture, which, acting as a mucolytic agent, reduces the viscosity of the fluid and stimulates natural drainage [37]. The presence of a biomolecule able to permeate a viscous fluid could favor the reduction of viscosity in bacterial biofilms associated with different diseases.

With the aim of testing the antimicrobial activity of flogomicina, a preliminary antimicrobial assay was carried out, using a modified disc diffusion test conducted on an MHA plate spread with *S. aureus* and *P. aeruginosa*. The first is a human ubiquitous Gram-positive bacteria responsible for outbreaks of nosocomial infections and, by attaching various host surfaces, it is able to form a biofilm which is difficult to eradicate with common treatments [38]. *P. aeruginosa* is a Gram-negative bacillus that most commonly affects the lower respiratory system in humans, and it commonly forms biofilm on medical devices and catheters [39]. Interestingly, flogomicina showed an inhibition diameter greater than pure NAC (Figure 2), which was particularly evident against *S. aureus*, suggesting that the formulation exhibits increased antimicrobial activity, underlined by the disk diffusion’s mean diameter. Moreover, in the biofilm assay (Figure 3) performed on *S. aureus*, we noted an increase in the number of non-viable cells in the NAC-treated wells at higher concentrations; conversely, flogomicina showed antibiofilm activity at all concentrations except for the minimum tested.

This effect could be attributed to the other components beyond NAC present in the mixture. The mechanism of the antimicrobial activity of bromelain is not well known; however, hydrolyzing the peptide and glycosidic bonds present within the glycoproteins and proteins in the biofilm and at the same time hydrolyzing complex carbohydrates may inhibit bacterial growth [15]. Moreover, resveratrol, an essential component in red wine, thanks to the inhibition of the hemolytic activity reported on S. aureus, could further reduce biofilm production [24].

Recently, several studies have demonstrated that in in vivo conditions, local coagulation induced by *S. aureus* favors fibrin deposits acting as a central structural component of the biofilm matrix [40,41]. In this work, we have demonstrated the higher fibrinolytic activity of flogomicina compared to plasmin and bromelain, highlighting the possibility of using a fibrinolytic formulation as a potentially novel therapeutic formulation to digest the biofilm matrix. It is likely that the complexity of the formulation increases the activity of bromelain, an effective fibrinolytic agent, since it stimulates the conversion of plasminogen to plasmin and subsequently causes fibrinolysis due to fibrin degradation [42].

In order to investigate the potential bacteriostatic or bactericidal effect of the formulation on *S. aureus*, a time-killing assay was performed. In Figure 5, no combinations resulted in a reduction in CFU to 0 or bactericide, but some mixtures showed more inhibition of growth bacteria than others at the final time point of five days. Specifically, pelargonium, ascorbic acid, and bromelain mixture and NAC alone revealed a higher reduction in CFU concerning the control, but this was lower than flogomicina, which exhibited bactericidal activity 20% greater than treatment with NAC alone and 55% greater than the mixture. These results demonstrated that the components present in flogomicina enhance the antibacterial activity of the NAC. Terlizzi et al. reported that pelargonium combined with NAC has a strong and long-term bacteriostatic activity related to its anti-oxidant activity [26]. Moreover, it was reported that ascorbic acid had antibacterial activity against opportunist bacteria, enhancing the antimicrobial activity of different antibacterial agents [18]. Additionally, bromelain helps neutralize bacterial activity thanks to its proteolytic enzyme that alters the membrane permeability, principally the peptidoglycan cell wall, thus resulting in mortal damage to the cells [43].

Since it has been reported that NAC has a positive modulatory effect when used in combination with different antibiotics [44,45], to expand the knowledge base on combined treatments, we performed a time-killing assay, analyzing the effect of flogomicina containing NAC with amoxicillin, an antibiotic commonly used in the primary care setting and covering an extensive range of Gram-positive bacteria [46]. Figure 6 shows how the administration of flogomicina three times a day in combination with amoxicillin can significantly reduce bacterial vitality more than flogomicina or amoxicillin alone, promoting a reduction of the antibiotic dosage over time, after only 14 days.

In order to broaden our studies on other bacteria responsible for chronic infections such as prostatitis or catheter-associated infections, a time-killing assay on *E. coli* and *P. mirabilis*, a common uropathogenic Gram-negative bacteria [47,48,49], was accomplished. On these pathogens, our results (Figure 6 and Figure 7) show that flogomicina attenuates the growth of bacteria more than only NAC. These data could be explained by the presence of the other component in the mixture, which has shown a strong activity on different urinary pathogens. Indeed, Vitamin C exercised an inhibitory effect on urinary pathogens *E. coli* and *K. pneumoniae* [50], and resveratrol reduced swarming motility in *P. mirabilis* [51], and uropathogenic *E. coli* [23].

Generally, the adhesion of bacteria to different types of surfaces entails a complex interplay of physical, chemical, and biological factors [52]. It is interesting to note that flogomicina reduced the adhesion of different Gram-negative bacteria to abiotic surfaces more than NAC, showing an anti-adhesive effect (Figure 8). This activity could be caused by the fact that NAC degrading relevant disulfide bridges of bacterial adhesins can prevent specific bacteria adhesion. On the other hand, in the formulation, the combination of more substances with NAC could change its free available groups, further reducing the bacterial adhesion [34].

## 5. Conclusions

This work represents a starting point for the evaluation of possible therapeutic applications of the natural multiple-component mixture of natural ingredients named flogomicina as an antimicrobial and antibiofilm agent. This novel mixture, made up of NAC and different natural ingredients (bromelain, ascorbic acid, Ribes nigrum, resveratrol, and pelargonium), has shown the ability to promote in vitro NAC permeation, fibrinolytic, antiadhesion, and antibiofilm activity against both Gram+ and Gram- bacteria (*S. aureus*, *E. coli*, *P. mirabilis,* and *P. aeruginosa*). These results highlight that the mixture of natural antioxidants with NAC does not decrease its antibacterial activity, but also increases its effectiveness against different bacteria. Moreover, flogomicina, increasing the antibacterial activity of amoxicillin, offers a safe and natural way of restricting the growth of bacteria by reducing the dosage of the antibiotics and consequently, their resistance. Based on these in vitro results, further investigations are needed to understand the mechanism of action of NAC and how it may interact with the other components of the mixture and other antibacterial agents, i.e., amoxicillin, that could promote its potential clinical application.

## Figures and Tables

**Figure 1 life-13-01005-f001:**
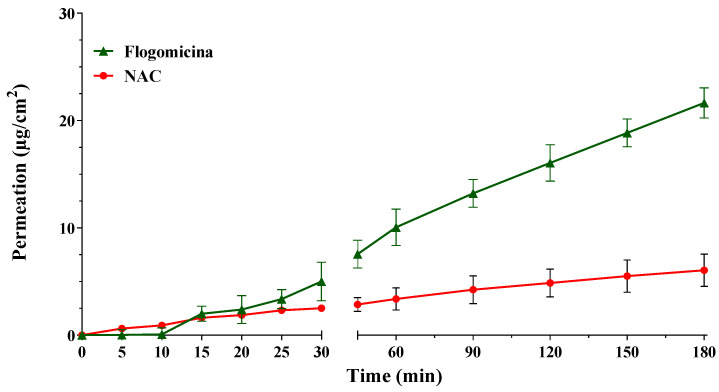
Permeation profile of NAC and NAC released from flogomicina through the artificial fluid. Media ± SD; (*n* = 6).

**Figure 2 life-13-01005-f002:**
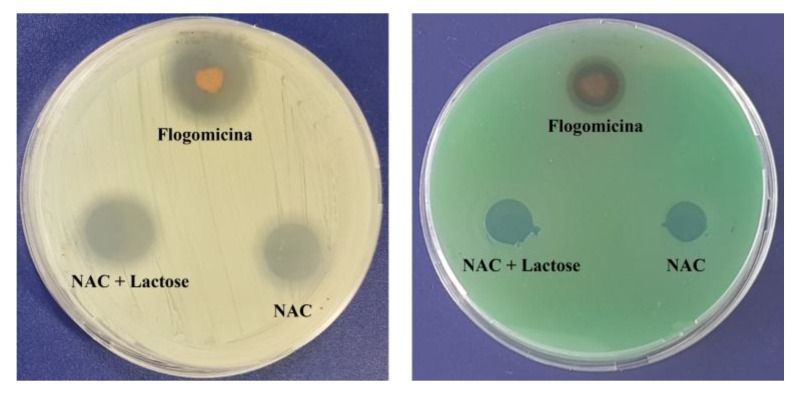
Antimicrobial activity against *S. aureus* (**left**) and *P. aeruginosa* (**right**) after 24 h, using a diffusion test. Flogomicina was applied in concentrations corresponding to 5 mg of NAC raw material. Pure NAC and NAC diluted with lactose were used as controls.

**Figure 3 life-13-01005-f003:**
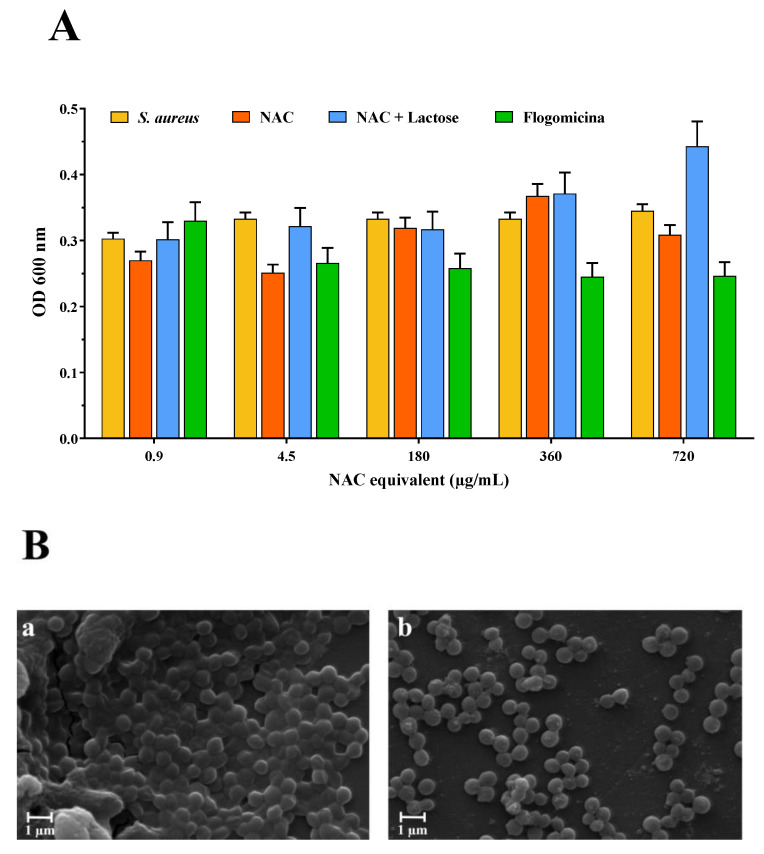
Anti-biofilm activity of flogomicina compared to pure NAC and NAC diluted with lactose. Panel (**A**): anti-biofilm activity after 48 h against *S. aureus* using optical density (OD) at 600 nm, expressed as mean ± SD; (*n* = 3); Panel (**B**): SEM microphotographs of *S. aureus* biofilm before (**a**), and after treatment with flogomicina at 360 µg/mL equivalent of NAC (**b**).

**Figure 4 life-13-01005-f004:**
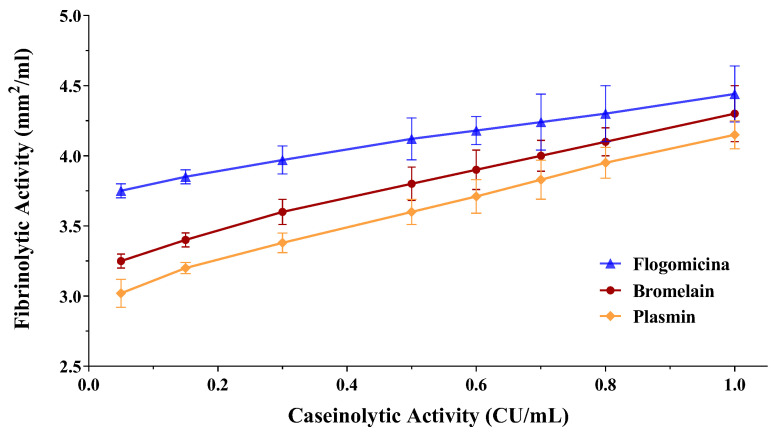
Fibrinolytic activity of human plasmin (used as control), bromelain, and flogomicina measured by a fibrin plate assay at 37 °C for 12 h. A double-logarithmic relationship between fibrinolytic activity (mm^2^) and caseinolytic activity (CU) is represented, and values are expressed as mean ± SD; (*n* = 3).

**Figure 5 life-13-01005-f005:**
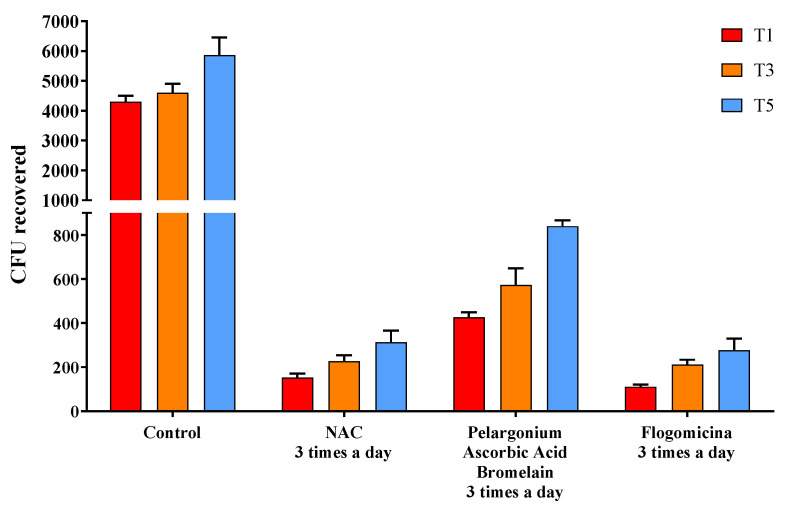
Antimicrobial activity against *S. aureus* obtained by a time-killing assay at 1, 3, and 5 days of incubation. Treatment with flogomicina in comparison with NAC, and the mixture containing pelargonium, ascorbic acid, and bromelain. A saline solution was used as a control. The *x*-axis represents CFU, or colony-forming units. Mean ± SD; (*n* = 6).

**Figure 6 life-13-01005-f006:**
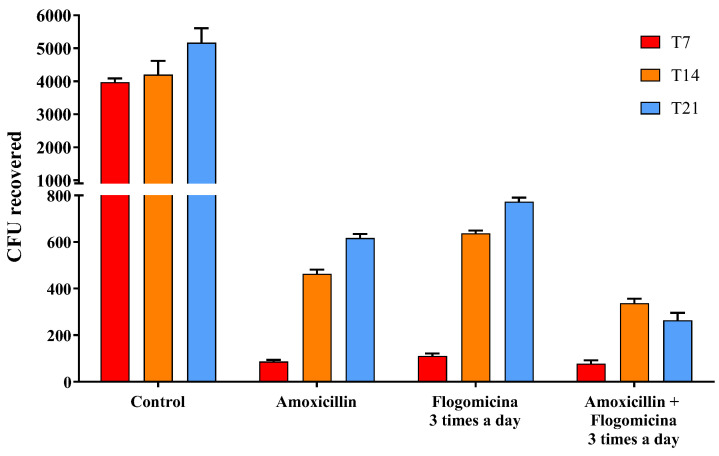
Antimicrobial activity of amoxicillin, flogomicina, and a mixture of amoxicillin and flogomicina alone against *S. aureus* by time-killing tests at 7-, 14-, and 21-day incubation. A saline solution was used as a control. The *x*-axis represents CFU, or colony-forming units. Mean ± SD; (*n* = 6).

**Figure 7 life-13-01005-f007:**
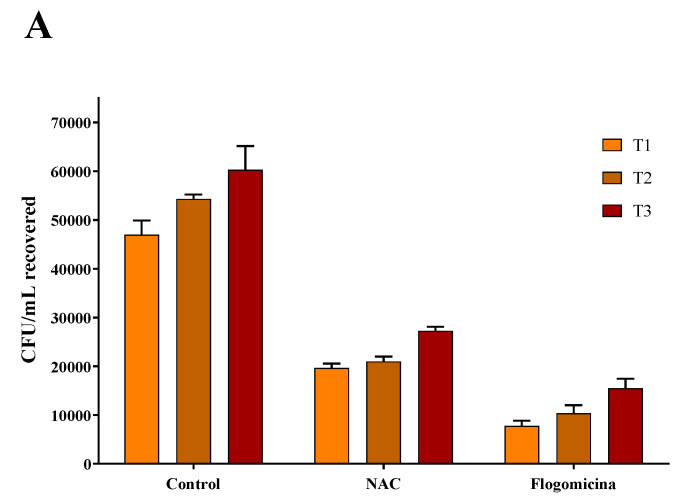
Antimicrobial activity of flogomicina and NAC against *E. coli* (Panel **A**) and *P. mirabilis* (Panel **B**) by time-killing tests at 1, 2, and 3 days incubation. A saline solution was used as a control. The *x*-axis represents CFU/mL. Mean ± SD; (*n* = 6).

**Figure 8 life-13-01005-f008:**
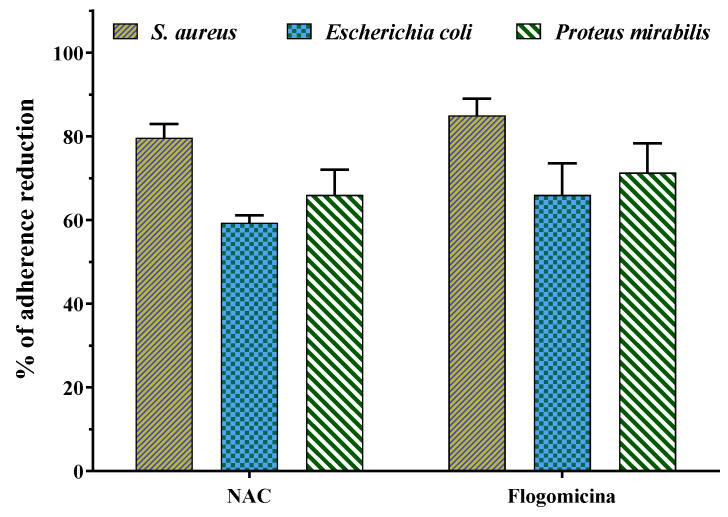
Effect of NAC alone and flogomicina mixture on the adherence of the tested microorganisms *S. aureus*, *E.coli*, and *P. mirabilis*; values are expressed as mean ± SD; (*n* = 3).

**Table 1 life-13-01005-t001:** Flogomicina mixture composition.

Ingredient	Amount (mg)	Concentration (*w*/*w*)
N-acetylcysteine	300	37.8
Bromelain	200	25.2
Ascorbic acid	100	12.6
Ribes nigrum L.	100	12.6
Resveratrol	50	6.3
Pelargonium	45	5.5

## Data Availability

Not applicable.

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
