# Peer review of "Flogomicina: A Natural Antioxidant Mixture as an Alternative Strategy to Reduce Biofilm Formation"

_life, 2023, doi:10.3390/life13041005_

Round 1

Reviewer 1 Report

This research was discussed the antibacterial activity by biofilm, the topic was novelty, but I suggest some comments as follow:

1.The author should describe the amin of this designed formulation for biofilm, especial for table 1 and in introduction section.

2.The author should try to describe the antibactial mechanism for this biofile, especially the Gram positive and Gram negative must be different.

3.The basic properties of this biofilm should be described, i.e. SEM or TEM?

Author Response

The authors appreciate the suggestion of the reviewer, thus, to improve the quality of the work, the manuscript has been edited.

This research was discussed the antibacterial activity by biofilm, the topic was novelty, but I suggest some comments as follow:

1.The author should describe the amin of this designed formulation for biofilm, especial for table 1 and in introduction section.

The text has been edited to better explain the aim of the work.

Line 83 “Therefore, this work aimed to combine a pool of natural antioxidants in a novel mixture named flogomicina as an alternative strategy to reduce antibiotic resistance. In detail, flogomicina composed of NAC, bromelain, ascorbic acid, Ribes nigrum, resveratrol, and pelargonium was tested to evaluate its antimicrobial activity favoring the breakdown of the bacterial biofilm, and the reduction of the bacteria adhesion on abiotic surfaces”

Line 100 “After a series of pilot experiments, Flogomicina (302021000020324), belonging to the category of food supplements, was prepared by combining different natural components (as reported in Table 1) in a specific amount using a geometric dilution technique to obtain a homogeneous physical mixture

2.The author should try to describe the antibactial mechanism for this biofile, especially the Gram positive and Gram negative must be different.

In the discussion, we have reported a hypothetic mechanism of action of flogomicina for bacteria biofilm.

Line 366 “Moreover, Moreover, in the biofilm assay (Fig. 3) performed on S. aureus, we noted an increase in the number of non-viable cells in the NAC-treated wells at higher concentrations; differently, flogomicina showed antibiofilm activity at all concentrations except for the minimum tested. This effect could be attributed to the other components beyond NAC present in the mixture. The mechanism of the antimicrobial activity of bromelain is not well known, but, hydrolyzing the peptide and glycosidic bonds present within the glycoproteins and proteins in the biofilm and at the same time hydrolyzing complex carbohydrates, may inhibit bacterial growth. Moreover, resveratrol, an essential component in red wine, thanks to the inhibition of the hemolytic activity reported on S. aureus. could further reduce biofilm production”.

3.The basic properties of this biofilm should be described, i.e. SEM or TEM?

The text has been implemented with images and comments following the revisor's suggestion. 

Reviewer 2 Report

In the article proposed by Chiara Amanteand colleagues,  The role of a novel natural antioxidants pool in microbial 2 biofilm reduction , aiming to demonstrate Flogomicina as a novel mixture of natural components to be used as an antimicrobial and antibiofilm agent.

The authors describe the important of 6 antioxidants components that have antibiofilm effects and claims that a mixture of these six components had a significant effect as antibiofilm and antimicrobial which eventually help to reduce the bacteria’s resistance to antimicrobial agents.

Overall quality of this manuscript can be accepted and the finding is suitable to be published, but some minor modifications are required:

1-      The title of this article needs to be descriptive, direct.

Some suggested titled:

·       Flogomicina; natural antioxidants,  mixure to reduce biofilm formation.

·       Flogomicina as an alternative strategy to combat biofilm development.

·       Reduction biofilm formation using novel natural antioxidants mixture "Flogomicina"

2-      Last paragraph in the introduction section (line 81 to line 88) needs to be revised since it seems too wordy and unhelpful. Rewrite the paragraph so it can help the reader to get to the points easily.

3-      In Line 425, can you please explain how this mixture " Flogomicina"  can used as antimicrobial since it was claimed. 

Thank you  

Author Response

The authors appreciate the suggestion of the reviewer, thus, to improve the quality of the work, the manuscript has been edited.

1.The title of this article needs to be descriptive, direct.

Some suggested titled:

Flogomicina; natural antioxidants, mixure to reduce biofilm formation.

Flogomicina as an alternative strategy to combat biofilm development.

Reduction biofilm formation using novel natural antioxidants mixture "Flogomicina"

The authors appreciate the suggestion. The new proposal is:

“Flogomicina: a natural antioxidants mixture as an alternative strategy to reduce biofilm formation

2.Last paragraph in the introduction section (line 81 to line 88) needs to be revised since it seems too wordy and unhelpful. Rewrite the paragraph so it can help the reader to get to the points easily.

The authors agree with the suggestion of the reviewer, therefore the paragraph has been edited to make clearer the aim of this work

Line 83 “Therefore, this work aimed to combine a pool of natural antioxidants in a novel mixture named flogomicina as an alternative strategy to reduce antibiotic resistance. In detail, flogomicina composed of NAC, bromelain, ascorbic acid, Ribes nigrum, resveratrol, and pelargonium was tested to evaluate its antimicrobial activity favoring the breakdown of the bacterial biofilm, and the reduction of the bacteria adhesion on abiotic surfaces”.

3.In Line 425, can you please explain how this mixture " Flogomicina"  can used as antimicrobial since it was claimed. 

Line 425 “This novel mixture made by NAC and different natural ingredients (bromelain, ascorbic acid, Ribes nigrum, resveratrol, and pelargonium) has shown the capability ability to promote in vitro NAC permeation, fibrinolytic, antiadhesion, and antibiofilm activity against both Gram+ and Gram- bacteria (S. aureus, E. coli, P. mirabilis, and P. aeruginosa)”.

Reviewer 3 Report

The manuscript deals with the potential use of a mixture of N-acetyl cysteine and natural ingredients (Flogomicina) to reduce biofilm formation.

It is a descriptive study, well presented with a detailed work conducted.  As interesting points it could be highlighted the use of nontoxic natural ingredients. In fact, maybe the main point of the article is the potential synergism with antibiotics as amoxicillin.

The subject is interesting and the methodology was indicated to the item and has been conducted properly nevertheless, some corrections must be performed in the text.

I believe that the work is worth to be published after some major corrections.

Some comments to the authors are made over the pdf.

Author Response

The authors strongly appreciate the comments of the reviewer. Following the suggestion, in the text, the use of natural ingredients was highlighted as well as the synergism effect with amoxicillin. 

For simplicity and linearity, in the attached, it is possible to find the pdf with answers to the comments.

Reviewer 4 Report

1.      The abstract section must be rewritten, the aims of the study must be clarified, and some numerical data about the main results must be added.

2.      Line 156 to 0.8 108 cells/mL must be correct to 0.8×108 cells/mL.

3.      The Flogomicina do not need to be written in capitalization in all manuscript text.

4.      Line 162 Biofilm formation assay, this test does not explain how to calculate the reduction percentage of biofilm formation.

5.      Line 177 Each antimicrobial assay was performed in triplicated on separate days. This test is a biofilm formation assay, how is the antimicrobial assay?

6.      Abbreviations. When used for the first time you write complete and abbreviation between brackets? Line 181 NIH.

7.      The term strain is not italicized on line 188.

8.      Line 190 1.5 × CFU/mL must be corrected to 1.5 × 106 CFU/mL.

9.      Lines 234 and 235 with the aim to investigate the ability of pure NAC, in comparison with NAC released from Flogomicina, to break disulfide bridges between macromolecules, must be rephrased and deleted reference 34.

10.  Lines 268 and 269 Anti-biofilm activity of Flogomicina compared to pure NAC and NAC diluted with lactose 268 after 48 hours against S. aureus using optical density (OD) at 600 nm. Must be deleted using optical density (OD) at 600 nm.

11.  Line 286 E. Coli, word coli must be start with small letter.

12.   Lines 288 and 289 words Ribes, Resveratrol, Pelargonium, Ascorbic acid, and Bromelain do not need to be written in capitalization in all manuscript text.

13.  Figures 5, 6, and 7 are unclear and very difficult to understand; therefore, the authors must be clarified.

Author Response

1.The abstract section must be rewritten, the aims of the study must be clarified, and some numerical data about the main results must be added.

We understand the concerns of the referee. Therefore abstract has been following the revisor’s suggestion.

2.Line 156 to 0.8 108cells/mL must be correct to 0.8×10cells/mL.

Sorry for the typos. The correction has been made.

3.The Flogomicina do not need to be written in capitalization in all manuscript text.

The text was edited following the suggestion.

4.Line 162 Biofilm formation assay, this test does not explain how to calculate the reduction percentage of biofilm formation.

We understand the concerns of the referee. In the methods, the reduction percentage of biofilm is not reported because the graph reported the values of OD compared with respect to the control.

5.Line 177 Each antimicrobial assay was performed in triplicated on separate days. This test is a biofilm formation assay, how is the antimicrobial assay?

The authors apologize for the typos reported in the manuscript. The text has been corrected.

6.Abbreviation When used for the first time you write complete and abbreviation between brackets? Line 181 NIH.

In the text was added the complete indication of NIH that is the standard unit used for thrombin.

7.The term strain is not italicized on line 188.

The text was edited.

8.Line 190 1.5 × CFU/mL must be corrected to 1.5 × 106CFU/mL.

Sorry for the typos. The correction has been made.

9.Lines 234 and 235 with the aim to investigate the ability of pure NAC, in comparison with NAC released from Flogomicina, to break disulfide bridges between macromolecules, must be rephrased and deleted reference 34.

Thank you for the suggestion. The text has been edited in:

Line 242 “With the aim to investigate the ability of pure NAC, in comparison with NAC released from flogomicina, to permeate through an artificial fluid(AF), in vitro permeation studies were conducted using Franz-type vertical diffusion cells”.

10.Lines 268 and 269 Anti-biofilm activity of Flogomicina compared to pure NAC and NAC diluted with lactose 268 after 48 hours against  aureususing optical density (OD) at 600 nm. Must be deleted using optical density (OD) at 600 nm.

The suggestion has been followed.

11.Line 286 E. Coli, word coli must be start with small letter.

The mistake was corrected.

12.Lines 288 and 289 words Ribes, Resveratrol, Pelargonium, Ascorbic acid, and Bromelain do not need to be written in capitalization in all manuscript text.

The suggestion has been followed

13. Figures 5, 6, and 7 are unclear and very difficult to understand; therefore, the authors must be clarified.

In order to make clearer the results, text and images were modified and better explained.

Round 2

Reviewer 3 Report

Thanks for your corrections!!!

Reviewer 4 Report

Thank you for revising the manuscript according to my suggestions.